# Smokers’ and Nonsmokers’ Receptivity to Smoke-Free Policies and Pro- and Anti-Policy Messaging in Armenia and Georgia

**DOI:** 10.3390/ijerph17155527

**Published:** 2020-07-30

**Authors:** Marina Topuridze, Carla J. Berg, Ana Dekanosidze, Arevik Torosyan, Lilit Grigoryan, Alexander Bazarchyan, Zhanna Sargsyan, Varduhi Hayrumyan, Nino Maglakelidze, Lela Sturua, Regine Haardörfer, Michelle C. Kegler

**Affiliations:** 1Non-Communicable Diseases Department, National Center for Disease Control and Public Health, Tbilisi 0198, Georgia; topuridzemarina@gmail.com (M.T.); ani.dekanosidze@gmail.com (A.D.); nmaglakelidze@yahoo.com (N.M.); lela.sturua@ncdc.ge (L.S.); 2School of Natural Sciences and Medicine, Ilia State University, Tbilisi 0162, Georgia; 3Department of Prevention and Community Health, Milken Institute School of Public Health, George Washington Cancer Center, George Washington University, Washington, DC 20052, USA; 4National Institute of Health Named after Academician S. Avdalbekyan, MoH, Yerevan 0051, Armenia; torossianarevik@gmail.com (A.T.); lilitgri@yahoo.com (L.G.); bazarchyan@gmail.com (A.B.); 5Turpanjian School of Public Health, American University of Armenia, Yerevan 0019, Armenia; zhsargsyan@aua.am (Z.S.); vhayrumyan@aua.am (V.H.); 6Public Health Department Petre Shotadze Tbilisi Medical Academy, Tbilisi 0144, Georgia; 7Department of Behavioral, Social, and Health Education Sciences, Rollins School of Public Health, Emory University, Atlanta, GA 30322, USA; regine.haardoerfer@emory.edu (R.H.); mkegler@emory.edu (M.C.K.); 8Winship Cancer Institute, Emory University, Atlanta, GA 30322, USA

**Keywords:** tobacco control, policy, smoke-free air policy, secondhand smoke exposure

## Abstract

Garnering support for smoke-free policies is critical for their successful adoption, particularly in countries with high smoking prevalence, such as Armenia and Georgia. In 2018, we surveyed 1456 residents (ages 18–64) of 28 cities in Armenia (*n* = 705) and Georgia (*n* = 751). We examined support for cigarette and electronic nicotine delivery systems (ENDS)/heated tobacco product (HTP) smoke-free policies in various locations and persuasiveness of pro- and anti-policy messaging. Participants were an average age of 43.35, 60.5% female, and 27.3% current smokers. Nonsmokers versus smokers indicated greater policy support for cigarette and ENDS/HTP and greater persuasiveness of pro-policy messaging. Armenians versus Georgians generally perceived pro- and anti-policy messaging more persuasive. In multilevel linear regression, sociodemographics (e.g., female) and tobacco use characteristics (e.g., smoking less frequently, higher quitting importance) correlated with more policy support. Greatest policy support was for healthcare, religious, government, and workplace settings; public transport; schools; and vehicles carrying children. Least policy support was for bar/restaurant outdoor areas. The most compelling pro-policy message focused on the right to clean air; the most compelling anti-policy message focused on using nonsmoking sections. Specific settings may present challenges for advancing smoke-free policies. Messaging focusing on individual rights to clean air and health may garner support.

## 1. Introduction

Low- and middle-income countries (LMICs) are disproportionately affected by tobacco-related diseases and deaths [1] including those attributed to secondhand smoke exposure (SHSe) [2]. One high-risk region for tobacco use is the area of the former Soviet Union [3]. Armenia and Georgia are two LMICs in this region that represent among the highest smoking prevalence among men (11th and 6th highest in the world; 52.3% and 57.7%, respectively), albeit lower prevalence among women (1.5% and 5.7%, respectively) [4]. Moreover, recent data indicate high SHSe in these countries [5,6], even in places where smoking is banned [5]. For example, 2014 data indicated that 42.2% of Georgian adults reported daily SHSe, with past-week SHSe being 54.2% in the home, 29.9% in indoor public places, and 33.0% in outdoor public places [6]. In Armenia, the 2016–2017 STEPS data indicated that 26.6% and 56.4% of adults reported past 30-day SHSe in the home and workplace, respectively [7]. Moreover, alternative tobacco products, such as electronic nicotine delivery systems (ENDS) and heated tobacco products (HTPs), have become increasingly prevalent in Armenia and Georgia [1,8]. For example, 2017 data indicate that 13.2% of Georgian youth (17.3% of boys, 7.7% of girls) reported past 30-day ENDS use [9]; reliable estimates of use among adults in Armenia and Georgia are not available. These data underscore an urgent need reduce SHSe due to cigarettes and ENDS/HTPs in these countries.

The World Health Organization (WHO) Framework Convention on Tobacco Control (FCTC) mandates that nations that ratify the FCTC implement specific evidence-based tobacco control policies, including comprehensive public smoke-free legislation. Public smoke-free policies are effective in reducing SHSe, youth tobacco use, overall use prevalence, and tobacco-related morbidity and mortality [10], as well as promoting cessation and harm reduction in smokers [10]. The FCTC was ratified in 2004 and 2006 in Armenia and Georgia, respectively; however, few FCTC-recommended policies had been implemented until recently.

In 2004, Armenia banned the consumption of tobacco in educational, cultural, healthcare, public transportation, and other public settings except dining facilities (e.g., restaurants, bars). In February 2020, Armenia adopted new legislation so that existing tobacco policies apply to alternative tobacco products (e.g., e-cigarettes, hookah) and extends smoke-free policies to all public places (including cafes/restaurants). In Georgia, new progressive tobacco control laws were implemented in 2017–2018, including a comprehensive smoke-free air policy that similarly covers all alternative tobacco products across a broad range of indoor and outdoor areas.

While policy progress is promising, data indicate that compliance with such smoke-free policies is poor [5,11], perhaps due to limited policy support. This may be in part due to “top-down approaches” in which national governments might adopt policies without full consideration of public sentiment surrounding tobacco use and related policies. Favorable attitudes towards tobacco control can contribute to effective policy adoption [12], implementation [13], and tobacco-related attitude and behavior change [14,15].

Understanding groups that differentially support these policies can inform targeted campaigns to encourage support for and compliance with new policies [16,17]. This is particularly crucial as policy progress is being made and as policies targeting novel products, such as ENDS and HTPs, are being considered and/or implemented. Past research has found that nonsmokers (vs. smokers), women (vs. men), and older individuals are more supportive [18]; there have been mixed findings regarding educational attainment and policy support [18]. Despite this literature, little research has focused on countries such as Armenia and Georgia. A 2014 study documented similar findings in Georgia, such that nonsmokers, women, and those older showed more smoke-free policy support [6]; however, this study did not assess attitudes toward ENDS/HTP policies.

Public health efforts to shift social norms and build policy support and compliance have included various strategies, particularly media campaigns and advocacy [19,20]. A range of arguments related to the impact of such policies on health, economic issues, youth prevention, individual rights, and morality have been used both to bolster support and opposition to such policies [21,22]. Prior research examined differences in perceptions in policies and messaging strategies with regard to the values and cultures of different populations across the US [23,24]; southeasterners (vs. others) were more persuaded by pro-tobacco tax messaging involving hospitality and protecting youth [23] and pro-smoke-free policy messaging involving economic impact, religion/morality, and hospitality [24].

Future efforts to garner support for and compliance with smoke-free policies in Armenia and Georgia could involve public health campaigns using messages reflecting the cultures and values of these countries. Given the sociopolitical histories in Armenia and Georgia and the prominence of religion and family in these cultures [25,26], messages appealing to ideals of individual freedom, rights, and responsibilities; fellowship and hospitality; religion and morality; and/or protecting youth might be particularly effective in this region. Moreover, such campaigns could involve messages aimed at addressing individuals’ concerns about such policies. One prior study in Georgia documented that, while SHSe was perceived to be dangerous, many individuals had misconceptions about how to protect others from SHSe (e.g., nonsmoking sections) and were concerned about smoke-free policies with regard to economic impact [27].

Given the gaps in the literature and the need to advance tobacco control in Armenia and Georgia, this study examined (1) support for smoke-free policies, both with regard to cigarettes and ENDS/HTPs, across various locations; (2) persuasiveness of messaging in support and opposition of such policies; and (3) correlates of these outcomes, using 2018 survey data from adults in 28 cities in Armenia and Georgia (14/country). Given the recent progressiveness of tobacco control in Georgia relative to Armenia, we anticipate greater support for such policies and persuasiveness of pro-policy messaging among Georgians versus Armenians. Additionally, we hypothesize that previously identified correlates of support for smoke-free policies and related messaging (e.g., female, nonsmoker) will hold in these populations, that policies in certain contexts (e.g., healthcare settings) will receive more support than in other settings (e.g., outdoor areas), and that some messaging strategies (e.g., health, individual rights) will be particularly persuasive. Given that we anticipate striking differences in policy support between smokers and nonsmokers, we first compare these groups in relation to our outcomes and then conduct our primary (multivariable) analyses among these groups separately.

## 2. Materials and Methods

### 2.1. Ongoing Study Overview

This study was approved by the Institutional Review Boards of Emory University (#IRB00097093), the National Academy of Sciences of the Republic of Armenia (#IRB00004079), the American University of Armenia (#AUA-2017–013), and the National Center for Disease Control and Public Health of Georgia (#IRB00002150). The ongoing parent study is more fully described elsewhere [28] and briefly described here. This study uses a matched-pairs community-randomized controlled trial to examine the effectiveness of local coalitions in promoting smoke-free air in Armenia and Georgia. We purposively selected 14 “communities” (i.e., municipalities) per country with small to medium populations. Communities were paired in each country based on region (and distance from Yerevan or Tbilisi), population size, and local public health branch/center budget, then randomly assigned to intervention versus control conditions.

### 2.2. Data Collection

Among all 28 intervention and control communities, population-level surveys (i.e., of community member) were conducted before the launch of the coalition member trainings (October–November 2018) and then will be conducted at the culmination of coalition activity. Current analyses focus on baseline population-level surveys conducted in October–November 2018. We aimed to complete 50 surveys of eligible participants (i.e., ≥18 years old) in each community. Sampling strategies were different in the two countries because of availability of household data in Armenia (but not in Georgia) and the utility of “clusters” (i.e., geographically defined areas of 150 households) in Georgia (but not in Armenia). In both countries, we obtained census data for all households within the municipality limits from the Bureau of Statistics. In each household, the KISH method [29] was used to identify target participants, who we approached in person at their homes, provided a study description, taken through informed consent, and administered the survey via electronic tablets.

In Armenia, addresses in each city were randomly ordered; assessments began at the beginning of the list and continued until the target recruitment in each city (*n* = 50) was reached. Overall, 1128 households were visited, of which 27.4% (*n* = 309) were ineligible (9.3% no household member eligible, 10.6% closed door/not home/do not live there anymore, 6.6% non-existing address). Among the 819 eligible, 705 (86.1%) participated.

In Georgia, multistage cluster sampling was used to select study participants. In step 1, five clusters per city were identified. In step 2, 15 households per cluster were selected using a random walking method: the total number of households was divided by * 15 * (assuming ~75% response rate) to determine how many households needed to be skipped before arriving at the next designated household (e.g., if the municipality included 150 households, the data collector would go from the first selected household to the 10th). Overall, 958 households were visited, of which 5.0% (*n* = 48) were ineligible (no household member reachable or eligible). Among the 910 eligible, 751 (82.5%) participated.

### 2.3. Measures

Sociodemographics. Current analyses included age, sex, education level, employment status, monthly household income, marital status, and children under the age of 18 in the home.

Smoking Characteristics. We assessed lifetime use of cigarettes (as well as ENDS and HTPs) among all participants; among lifetime users, we assessed past 30-day use. Among past 30-day smokers, we assessed number of days smoked, cigarettes smoked per day (CPD), readiness to quit in the next 30 days and 6 months, and number of quit attempts in the past year (categorized as any vs. none). We also assessed the importance of quitting (“On a scale of 0 to 10, with 0 being not at all and 10 being extremely, how important is quitting smoking to you?”); a parallel item was used to assess confidence in quitting.

Reactions to Smoke-Free Policy. We asked, “To what extent do you support or oppose a complete cigarette smoking ban in the following settings?” and “To what extent do you support or oppose a ban on using e-cigarettes or heat-not-burn tobacco such as IQOS in the following settings?” The settings assessed in relation to cigarette smoking are listed in Table 2, with a subset (indicated in Figure 1) also being assessed in relation to ENDS/HTPs. The response options included: Strongly oppose, somewhat oppose, Somewhat support, Strongly support, (Don’t know), and (Refuse). Given a sizeable proportion of participants indicating responses of “don’t know” (6 with 5–10% indicating “don’t know”; 10 with 10–15% indicating “don’t know”), “don’t know” was coded as the middle response (i.e., 1 = Strongly oppose, 2 = Somewhat oppose, 3 = Don’t know, 4 = Somewhat support, 5 = Strongly support). Composite index scores were created for items regarding cigarettes, ENDS/HTPs, and both by totaling the responses and dividing by the number of items in each index. Cronbach’s alpha for responses related to cigarettes, ENDS/HTPs, and both were 0.91, 0.93, and 0.93, respectively. The correlation between the cigarette index score and ENDS/HTPs index score was 0.46 (*p* < 0.001).

Persuasiveness of Policy-Related Messaging. We asked, “For each of the following statements, indicate the extent to which each statement is convincing or persuasive. We are not asking if you agree with them, but rather if you feel that the information given would convince or persuade you to support or oppose a smoke-free air policy.” The items are listed in Figure 2, and response options included: Not at all persuasive, moderately persuasive, extremely persuasive, (Don’t know), and (Refuse). Given a sizeable proportion of participants indicating responses of “don’t know” (3 with 5–10% indicating “don’t know”; 4 with 10–15% indicating “don’t know”), “don’t know” was coded as a middle response (i.e., 1 = Not at all persuasive, 2 = Don’t know, 3 = Moderately persuasive, 4 = Extremely persuasive). A composite index score was created for persuasiveness ratings for pro- and anti-policy messages by totaling the responses and dividing by the number of items in each index, respectively. Cronbach’s alpha for responses related to pro-policy messaging and anti-policy messaging were 0.78 and 0.86, respectively. The correlation between the index scores for pro-policy messaging and anti-policy messaging was 0.18 (*p* < 0.001).

### 2.4. Data Analysis

We first conducted descriptive analyses to characterize participants. Then, we conducted bivariate analyses to examine differences between smokers and nonsmokers in relation to (a) sociodemographics, (b) support for smoke-free policies in various settings, and (c) perceived persuasiveness of pro-policy and anti-policy messaging, respectively.

Using as outcomes the index scores created to summarize support for bans on cigarettes and bans on END/HTPs, respectively, we conducted smoking status stratified analyses. We used random intercept multilevel linear regression analyses accounting for the random effect of municipality using GenLinMixed with an identity link function in SPSS. We first ran unconditional models to estimate the unconditional intra-class correlation (ICC) and then expanded the full model, by including the random effect of municipality and by adding fixed effects for country, age, sex, employment status, relationship status, and children in the home together in one step. Additionally, analyses included fixed effects for tobacco use characteristics (i.e., former smoker status among nonsmokers; smoking frequency and quitting importance/confidence among smokers). We modeled an unstructured covariance matrix and excluded cases with missing data on covariates (ranging from 3–5%). Full Model ICCs were calculated for all full models. Similarly, we conducted multilevel linear regression analyses using as outcomes index scores for perceived persuasiveness of pro- and anti-policy messaging, respectively. All analyses were conducted in SPSS v. 26 (IBM, Armonk, New York, USA), and alpha was set at 0.05.

## 3. Results

### 3.1. Participant Characteristics

Participants were an average age of 43.35 years old, 60.5% female, 32.1% with a college education, and 49.0% employed (Table 1). Overall, 27.3% reported to smoking on some days or every day. Only 3.3% and 1.0% indicated lifetime use of ENDS and HTPs, respectively, with past 30-day use showing extremely low prevalence (ENDS, *n* = 9; HTPs, *n* = 4). Among smokers, 83.4% were not ready to quit in the next six months and 43.7% never tried to quit. Former smokers represented 9.6% of current nonsmokers. Compared to smokers, nonsmokers were more supportive of public policies banning cigarette and ENDS/HTPs use and reported pro-policy messaging as more persuasive (*p*’s < 0.001).

Across settings, nonsmokers reported greater support for both cigarette and ENDS/HTPs smoke-free policies (*p*’s < 0.05; Table 2). The support for cigarette smoke-free policy across all settings was relatively higher compared to ENDS/HTPs policy (Figure 1).

The greatest support (ave. > 4/5) was for policies in healthcare, religious, government, and workplace settings; public transport; schools; and vehicles with children present. The least support (ave. < 3/5) was for policies in outdoor areas of bars or restaurants. Support was mixed (ave. 3–4/5)—and higher in nonsmokers versus smokers (ave. > 1 point)—regarding indoor and outdoor areas of bars or restaurants, multiunit housing (common indoor/outdoor areas, individual units), and outdoor public areas (e.g., playgrounds, public transportation stops).

There were also differences across countries in relation to support for cigarette and ENDS/HTPs smoke-free policies (Table 2). Participants from Georgia versus Armenia reported greater support for cigarette smoke-free policies in healthcare facilities; religious institutions; schoolyards; outdoor areas of university or college campuses; bars, pubs and nightclubs; public transportation; and near public building entrances. However, participants from Armenia versus Georgia reported greater support for cigarette smoke-free policies in indoor areas of universities, institutes, colleges; outdoor terrace of restaurants, cafes, bars, and nightclubs; individual apt/condo units; public transportation stops and taxis; private vehicles with children present; and outdoor areas (e.g., playgrounds, parks, beaches, open stadiums). Finally, Georgian versus Armenian participants reported higher support for ENDS/HTP-related policies in universities/colleges indoor places.

In multilevel linear regression (Table 3a), among nonsmokers, correlates indicating greater support for cigarette smoke-free air policies included being female (B = 0.19, CI: 0.08, 0.23, *p* < 0.001) and not being former smokers (B = −0.21, CI: −0.35, −0.08, *p* = 0.002); there were no significant correlates of ENDS/HTP-related policy support. Among smokers, correlates of greater support for cigarette smoke-free air policies included smoking less than daily (vs. daily; B = 0.36, CI: 0.14, 0.57, *p* = 0.001) and reporting quitting as more important (B = 0.07, CI: 0.05, 0.09, *p* < 0.001); greater support for ENDS/HTPs policies was correlated with participants’ reporting quitting as more important (B = 0.03, CI: 0.00, 0.06, *p* = 0.047).

### 3.2. Persuasiveness of Messaging Strategies

Each pro-policy message was perceived as more persuasive among nonsmokers versus smokers (p’s < 0.001; Figure 2). The most compelling messaging strategy among nonsmokers and smokers focused on the right to breathe clean air (M ± SD: 3.67 ± 0.61 vs. 3.37 ± 0.85, *p* < 0.001), followed by SHSe health consequences (M ± SD: 3.59 ± 0.68 vs. 3.15 ± 0.96, *p* < 0.001); the least compelling messages focused on negligible policy impact on businesses (M ± SD: 2.85 ± 1.04 vs. 2.47 ± 1.07, *p* < 0.001). The most compelling anti-policy messaging focused on using smoking/nonsmoking sections (M ± SD: 2.76 ± 1.17 vs. 2.80 ± 1.13, *p* = 0.610), followed by consumers’ responsibility to guard against SHSe (M ± SD: 2.41 ± 1.19 v. 2.52 ± 1.16, *p* = 0.034); the least compelling was negative impact on businesses (M ± SD: 2.09 ± 1.14 v. 2.23 ± 1.12, *p* = 0.293).

In multilevel linear regression analyses (Table 3b), among nonsmokers, there were no significant correlates of perceiving pro-policy messaging as more persuasive; the only correlate of rating anti-policy messaging as more persuasive was living in Armenia (B = −0.95, CI: −1.21, −0.69, *p* < 0.001). Among smokers, correlates of rating pro-policy messaging as more persuasive included being Armenian (B = −0.46, CI: −0.66, −0.27, *p* < 0.001), being younger (B = −0.01, CI: −0.01, −0.002, *p* = 0.026), smoking less than daily (vs. daily; B = 0.31, CI: 0.04, 0.58, *p* = 0.022), and reporting quitting as more important (B = 0.05, CI: 0.02, 0.07, *p* < 0.000). Correlates of rating anti-policy messaging as more persuasive among smokers included being Armenia (B = −0.82, CI: −1.06, −0.58, *p* < 0.001) and smoking less than daily (vs. daily; B = 0.51, CI: 0.21, 0.81, *p* = 0.001).

## 4. Discussion

This study supported some hypotheses (e.g., sociodemographic and tobacco use factors associated with policy support and perceived persuasiveness of messaging), but not others. Notably, while Armenians versus Georgians reported less support for policies (on average) in several settings, country of residence was not a significant correlate of policy support in multivariable analysis. In addition, Armenians versus Georgians generally indicated that both pro- and anti-policy messaging was more persuasive. Below, we further discuss, contextualize, and interpret these and other findings.

Current results indicate high levels of support for cigarette smoke-free policies in Armenia and Georgia, particularly for policies covering healthcare, religious, government, and workplace settings; public transport; schools; and vehicles carrying children. Given the high level of support for cigarette smoke-free policies, it is important to consider why tobacco control policies, specifically comprehensive smoke-free policies, have lagged in Armenia and Georgia. One explanation might be that constituents are insufficiently engaged with lawmakers, which is critical in advancing tobacco control legislation whether the influence comes from the public health community or the tobacco industry [30]. Another explanation may stem from policymakers’ misconceptions about the negative health impacts of SHSe or the economic and public health benefits of smoke-free policies [12,31,32]. These misconceptions can be addressed easily, as the literature is clear on the health impact of SHSe and health benefits of smoke-free policies [10] and largely indicates neutral or positive impact of such policies on businesses (with findings otherwise generally produced by the tobacco industry) [10]. Finally, policymakers’ decisions may be influenced less by constituent sentiment and more by personal attitudes and interests [33,34], perhaps influenced by tobacco industry role in the economy and culture of many countries, including Armenia and Georgia [35,36].

Several previously identified factors associated with support for tobacco control were documented with relation to cigarette smoke-free air policies, for example, being female, nonsmokers, or smokers who less frequently smoke and perceive quitting as more important [18]. However, fewer were found with relation to policies covering ENDS/HTPs, perhaps due to fewer people being aware of these products, the literature being less mature regarding their health risks, and limited public health campaigns communicating the risks of such products [1,8,37,38].

Other important findings include the messaging strategies that might be most effective with these populations. Overall, pro-policy messaging strategies that were perceived as most persuasive were focused on individual rights to smoke-free air and negative SHSe health impact. The most compelling anti-policy message focused on using nonsmoking sections or consumer responsibility to guard against SHSe. These findings are similar to prior research in the US indicating the most effective pro-policy messages pertained to hospitality, health, and individual rights/responsibilities; the most persuasive anti-policy messages involved individual rights/responsibilities [24]. Moreover, current findings indicated that the least compelling messages in support and opposition focused on impact on businesses; prior research in the US also indicated that pro- and anti-policy messages focusing on economic impact were not particularly compelling [24].

Not surprisingly, those younger and smokers who smoked less frequently and reported quitting as more important were perceived pro-policy messaging as more persuasive [18]. Interestingly, Armenians were more likely to perceive both pro- and anti-policy messaging as more persuasive; this may be related to the fact that Georgia had (and has) made more progress in advancing smoke-free policies, which involved public health campaigns to garner such support. Perhaps the information provided to Armenians was perceived as more novel and thus more persuasive [39], given a relative dearth of such public health campaigns.

Current findings have important implications for research and practice. Research should examine the processes impeding the adoption of comprehensive smoke-free policies in countries like Armenia and Georgia, particularly given their sociopolitical histories and high tobacco use prevalence among men. Relatedly, studying how community engagement and coalition building can advance smoke-free policy legislation and use various messaging strategies to do so warrants research [30]. Given high levels of support for smoke-free policies and evidence for promising strategies for garnering support, smoke-free policy advocacy efforts should focus campaign messaging on the positive health impact of smoke-free policies and such policies protecting vulnerable populations. Collectively, these findings provide a foundation to inform the activities of public health practitioners to further the agenda of public smoke-free policy adoption.

This study has limitations. This sample may not represent the general adult populations of these countries. Additionally, the sampling/recruitment methods across countries differed by necessity and yielded different response rates and composition by sex and smoking status. Our results could also be biased due to several factors, such as unmeasured variables associated with differential participation. Finally, the cross-sectional nature and self-reported assessments limit the ability to make causal attributions or account for bias. Thus, these results must be cautiously interpreted.

## 5. Conclusions

Armenians and Georgians are highly supportive of cigarette smoke-free air policies and in healthcare, religious, government, and workplace settings; public transportation; schools; and vehicles carrying children. However, there was less support for such policies regarding other locations, particularly outdoor areas of bars and restaurants, and regarding ENDS/HTPs. This underscores the need to address what is known about the risks of secondhand exposure to the byproducts of ENDS/HTPs and to SHSe in a broad range of settings, including outdoor settings. Moreover, campaigns to garner support for smoke-free policies should emphasize individual rights to clean air and health and combat using nonsmoking sections by highlighting the relative ineffectiveness of such strategies.

## Figures and Tables

**Figure 1 ijerph-17-05527-f001:**
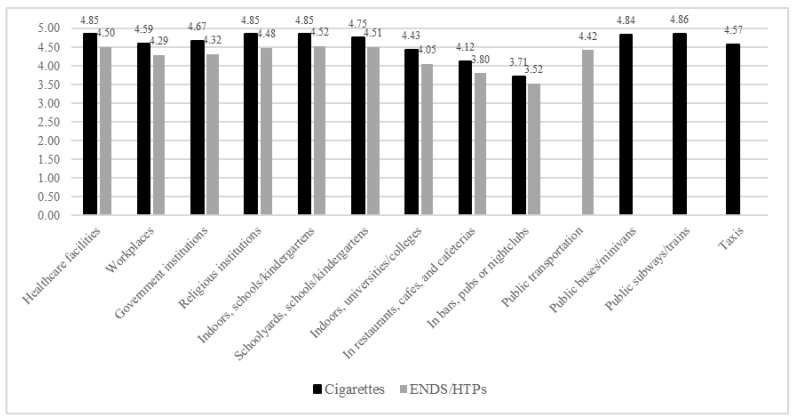
Support for cigarette smoking bans and bans on use of ENDS/HTPs in select settings. 1 = Strongly oppose, 2 = Somewhat oppose, 3 = Don’t know, 4 = Somewhat support, 5 = Strongly support. Note: All comparisons of cigarette vs. ENDS/HTPs policy support indicated significant differences.

**Figure 2 ijerph-17-05527-f002:**
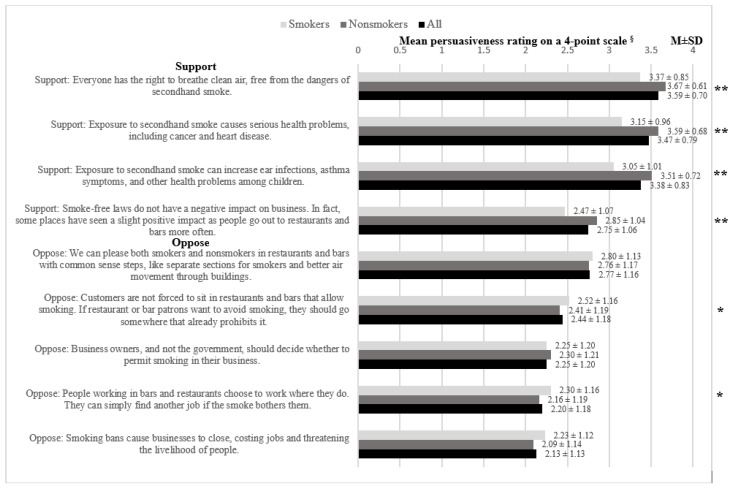
Nonsmokers’ and smokers’ rated persuasiveness of smoke-free policy messages. **^§^** 1 = Not at all, 2 = Don’t know, 3 = Moderately, 4 = Extremely; ** *p* < 0.001; * *p* < 0.05.

**Table 1 ijerph-17-05527-t001:** Participant characteristics and bivariate comparisons of nonsmokers’ vs. smokers’ support for smoke-free air policies.

Variable	All Participants, N = 1456 *n* (%) or M (SD)	Nonsmokers, *n* = 1058 (72.7%) *n* (%) or M (SD)	Smokers, *n* = 398 (27.3%) *n* (%) or M (SD)	*p*
Country, N (%)				<0.001
Armenia	705 (48.4)	561 (53.0)	144 (36.2)	
Georgia	751 (51.6)	497 (47.0)	254 (63.8)	
*Sociodemographics*				
Age, M (SD)	43.35 (13.49)	43.34 (13.59)	43.38 (13.26)	0.957
Sex, N (%)				<0.001
Male	575 (39.5)	207 (19.6)	368 (92.5)	
Female	881 (60.5)	851 (80.4)	30 (7.5)	
Education, N (%)				0.027
Less than high school	223 (15.3)	157 (14.8)	66 (16.6)	
High school	260 (17.9)	175 (16.5)	85 (21.4)	
Vocational school	407 (28.0)	307 (29.0)	100 (25.1)	
Some college	98 (6.7)	64 (6.0)	34 (8.5)	
College degree or more	468 (32.1)	355 (33.6)	113 (38.4)	
Employment, N (%)				<0.001
Employed	713 (49.0)	438 (41.4)	275 (69.1)	
Unemployed/other	743 (51.0)	620 (58.6)	123 (30.9)	
Income per month, N (%)				0.111
≤500 GEL/100,000 AMD	563 (45.1)	423 (47.7)	140 (42.3)	
>500 GEL/100,000 AMD	685 (54.9)	463 (52.3)	191 (57.7)	
Marital status, N (%)				0.050
Married/cohabitating	1061 (72.9)	784 (74.1)	277 (69.6)	
Other	395 (27.1)	274 (25.9)	121 (30.4)	
Children under 18 in the home, N (%)				0.097
No	702 (49.0)	497 (47.9)	205 (51.9)	
Yes	731 (51.0)	541 (52.1)	190 (48.1)	
*Smoking characteristics*				
Former smoker, N (%)				-
No	-	956 (90.4)	-	
Yes	-	102 (9.6)	-	
Number of days smoked, past 30, N (%)				-
Every day	-	-	350 (87.9)	
Some days	-	-	48 (12.1)	
CPD, M (SD)	-	-	21.27 (10.79)	-
Readiness to quit, next 6 months, N (%)				-
No	-	-	242 (83.4)	
Yes	-	-	48 (16.6)	
Lifetime quit attempt, N (%)				-
No	-	-	166 (43.7)	
Yes	-	-	214 (56.3)	
Importance of quitting, M (SD)	-	-	5.74 (3.23)	-
Confidence in quitting, M (SD)	-	-	4.79 (3.18)	-
*Support for bans index scores*, M (SD)				
Cigarettes	4.20 (0.66)	4.38 (0.56)	3.72 (0.66)	<0.001
ENDS/HTPs	4.24 (0.92)	4.34 (0.89)	3.99 (0.94)	<0.001
Both	4.21 (0.63)	4.37 (0.56)	3.80 (0.65)	<0.001
*Messaging persuasiveness index scores*, M (SD)				
Support	3.30 (0.66)	3.41 (0.57)	3.01 (0.79)	<0.001
Opposition	2.36 (0.94)	2.33 (0.93)	2.43 (0.94)	0.081

**Table 2 ijerph-17-05527-t002:** Nonsmokers and smokers support for smoke-free air policies in various settings (Means, Standard Deviations).

Cigarettes	All, N = 1456	Nonsmokers, *n* = 1058	Smokers, *n* = 398	*p*	Georgia, *n* = 751	Armenia, *n* = 705	*p*
Healthcare facilities	4.85 (0.62)	4.89 (0.54)	4.74 (0.78)	<0.001	4.89 (0.46) *	4.81 (0.76) *	0.019
Workplaces	4.59 (0.90)	4.76 (0.66)	4.11 (1.23)	<0.001	4.57 (0.91) **	4.60 (0.90) **	0.595
Government institutions	4.67 (0.78)	4.79 (0.61)	4.34 (1.03)	<0.001	4.66 (0.75) **	4.68 (0.81) **	0.612
Religious institutions	4.85 (0.60)	4.88 (0.53)	4.77 (0.75)	0.002	4.93 (0.35) **	4.77 (0.77) **	<0.001
Indoor areas of schools, kindergartens	4.85 (0.60)	4.88 (0.55)	4.79 (0.71)	0.018	4.86 (0.49) *	4.85 (0.69) *	0.772
Schoolyards of schools, kindergartens	4.75 (0.78)	4.77 (0.73)	4.66 (0.89)	0.015	4.87 (0.44) *	4.60 (1.00) ***	<0.001
Indoor areas of universities, institutes, colleges	4.43 (1.14)	4.56 (1.02)	4.09 (1.36)	<0.001	4.07 (1.33) *	4.81 (0.73) *	<0.001
In outdoor areas of university, college campuses	4.42 (1.09)	4.46 (1.05)	4.32 (1.21)	0.025	4.82 (0.53) *	4.00 (1.35) ***	<0.001
In restaurants, cafes, cafeterias	4.12 (1.30)	4.41 (1.04)	3.34 (1.55)	<0.001	4.17 (1.27) *	4.05 (1.32) ***	0.075
Outdoor terrace of restaurants, cafes, cafeterias	2.98 (1.60)	3.35 (1.53)	2.01 (1.35)	<0.001	2.78 (1.63) ***	3.19 (1.54) ***	<0.001
In bars, pubs, or nightclubs	3.71 (1.48)	4.02 (1.31)	2.90 (1.48)	<0.001	3.85 (1.43) *	3.57 (1.52) ***	<0.001
Outdoor terrace of bars, pubs, nightclubs	2.88 (1.60)	3.23 (1.54)	1.94 (1.32)	<0.001	2.72 (1.63) ***	3.04 (1.54) ***	<0.001
Indoor common areas of apts, condos	3.99 (1.38)	4.30 (1.16)	3.16 (1.56)	<0.001	3.98 (1.33) *	4.00 (1.43) ***	0.783
Outdoor common areas of apts, condos	3.76 (1.43)	4.06 (1.24)	2.96 (1.57)	<0.001	3.74 (1.43) ***	3.77 (1.42) ***	0.697
Within individual apt/condo units	3.79 (1.47)	4.10 (1.27)	3.00 (1.64)	<0.001	3.63 (1.52) ***	3.96 (1.39) ***	<0.001
Public bus or minivan stops	4.10 (1.29)	4.30 (1.12)	3.58 (1.54)	<0.001	4.03 (1.32) ***	4.18 (1.25) ***	0.023
In public buses, minivans	4.84 (0.62)	4.88 (0.54)	4.73 (0.78)	<0.001	4.85 (0.49) *	4.82 (0.73) *	0.367
Public subway, train stations	4.53 (0.96)	4.62 (0.85)	4.29 (1.19)	<0.001	4.62 (0.84) *	4.44 (1.08) *	<0.001
In public subways, trains	4.86 (0.60)	4.89 (0.53)	4.77 (0.75)	0.001	4.88 (0.45) *	4.83 (0.73) *	0.142
Taxis	4.57 (1.01)	4.77 (0.72)	4.05 (1.40)	<0.001	4.37 (1.15) ***	4.79 (0.78) ***	<0.001
Within 5 m public building entrances	3.33 (1.50)	3.62 (1.41)	2.56 (1.48)	<0.001	3.45 (1.39) ***	3.20 (1.61) ***	0.002
Private vehicles with children <18	4.75 (0.72)	4.83 (0.62)	4.55 (0.93)	<0.001	4.70 (0.68) ***	4.81 (0.77) ***	0.004
Playgrounds	4.37 (1.10)	4.55 (0.88)	3.89 (1.43)	<0.001	4.21 (1.18) **	4.54 (0.98) ***	<0.001
Parks, beaches	3.53 (1.47)	3.83 (1.35)	2.73 (1.49)	<0.001	3.21 (1.48) ***	3.88 (1.39) ***	<0.001
Other public outdoor areas (e.g., open stadiums)	3.41 (1.56)	3.72 (1.43)	2.59 (1.58)	<0.001	3.00 (1.61) ***	3.84 (1.38) ***	<0.001
**ENDS/HTPs**							
Healthcare facilities	4.50 (1.03)	4.53 (0.99)	4.42 (1.11)	0.077	4.49 (0.97) *	4.51 (1.09) ***	0.737
Workplaces	4.29 (1.13)	4.37 (1.07)	4.09 (1.26)	<0.001	4.29 (1.09) *	4.30 (1.19) ***	0.860
Government institutions	4.32 (1.09)	4.39 (1.02)	4.12 (1.22)	<0.001	4.30 (1.07) *	4.35 (1.11) ***	0.471
Religious institutions	4.47 (1.03)	4.50 (1.00)	4.41 (1.13)	0.174	4.53 (0.95) *	4.43 (1.12) ***	0.076
Indoors of schools, kindergartens	4.51 (1.00)	4.55 (0.95)	4.42 (1.13)	0.033	4.49 (0.99) *	4.55 (1.03) ***	0.238
Schoolyards of schools, kindergartens	4.51 (1.00)	4.53 (0.96)	4.44 (1.08)	0.131	4.50 (0.98) *	4.53 (1.03) ***	0.536
Indoors of universities, colleges	4.04 (1.34)	4.18 (1.24)	3.68 (1.52)	<0.001	3.63 (1.45) *	4.49 (1.05) ***	<0.001
In restaurants, cafes, cafeterias	3.80 (1.38)	4.05 (1.20)	3.13 (1.59)	<0.001	3.67 (1.37)	3.95 (1.38) ***	<0.001
In bars, pubs, nightclubs	3.51 (1.48)	3.77 (1.35)	2.84 (1.59)	<0.001	3.48 (1.45)	3.56 (1.52) ***	0.261
Public transportation	4.42 (1.08)	4.46 (1.03)	4.32 (1.18)	0.031	4.42 (1.04)	4.43 (1.13) ***	0.862

* Total ban implemented in the country; ** Partial restrictions implemented in the country; *** No restrictions implemented in the country.

**Table 3 ijerph-17-05527-t003:** Nonsmokers’ and smokers’ support for use bans of cigarettes and ENDS/HTPs and rated persuasiveness of smoke-free policy messages.

**(a) Nonsmokers’ and Smokers’ Support for Use Bans of Cigarettes and ENDS/HTPs**	**Nonsmokers**	**Smokers**
**Cigarettes, *n* = 1019**	**ENDS/HTPs, *n* = 1015**	**Cigarettes, *n* = 372**	**ENDS/HTPs, *n* = 376**
**B (95% CI)**	***p***	**B (95% CI)**	***p***	**B (95% CI)**	***p***	**B (95% CI)**	***p***
Intercept	4.14 (3.95, 4.33)	<0.001	4.38 (4.04, 4.71)	<0.001	3.31 (2.93, 3.68)	<0.001	3.97 (3.40, 4.54)	<0.001
Georgia (vs. Armenia)	0.11 (−0.06, 0.28)	0.198	0.00 (−0.32, 0.32)	0.977	−0.17 (−0.40, 0.06)	0.141	−0.18(−0.57, 0.20)	0.350
*Sociodemographics*								
Age	0.001 (−0.001, 0.001)	0.380	0.00 (0.00, 0.00)	0.507	0.00 (0.00, 0.01)	0.324	0.00 (−0.01, 0.01)	0.795
Female (vs. male)	0.19 (0.08, 0.23)	<0.001	0.02 (−0.15, 0.18)	0.845	0.13 (−0.12, 0.38))	0.309	0.07 (−0.31, 0.44)	0.730
Unemployed/other (vs. employed)	0.02 (−0.04, 0.08)	0.544	0.00(−0.10, 0.10)	0.982	−0.07 (−0.21, 0.06)	0.286	0.06(−0.15, 0.26)	0.589
Other (vs. married/cohabitating)	−0.01 (−0.09, 0.06)	0.714	−0.09 (−0.22, 0.03)	0.126	−0.04 (−0.20, 0.12)	0.619	−0.11 (−0.34, 0.13)	0.367
Children <18 in home (vs. no)	0.01 (−0.05, 0.08)	0.677	0.07 (−0.04, 0.18)	0.227	0.01 (−0.12, 0.15)	0.872	0.04 (−0.16, 0.24)	0.675
*Smoking Characteristics*								
Former smoker (vs. never)	−0.21 (−0.35, −0.08)	0.002	−0.03 (−0.24, 0.18)	0.763	-	-	-	-
Smoking some days (vs. every)	-	-	-	-	0.36 (0.14, 0.57)	0.001	0.23 (−0.08, 0.55)	0.151
Importance of quitting	-	-	-	-	0.07 (0.05, 0.09)	<0.001	0.03 (0.00, 0.06)	0.047
Confidence in quitting	-	-	-	-	−0.01 (−0.03, 0.01)	0.517	−0.01 (−0.04, 0.02)	0.494
Unconditional Model ICC	16.83%		19.58%		11.00%		21.18%	
Full Model ICC	15.79%		21.42%		15.31%		21.12%	
**(b) Nonsmokers’ and Smokers’ Rated Persuasiveness of Smoke-free Policy Messages**	**Nonsmokers**	**Smokers**
**Support, *n* = 1023**	**Opposition, *n* = 1016**	**Support, *n* = 375**	**Opposition, *n* = 373**
**B (95% CI)**	***p***	**B (95% CI)**	***p***	**B (95% CI)**	***p***	**B (95% CI)**	***p***
Intercept	3.31 (3.11, 3.52)	<0.001	2.66 (2.36, 2.95)	<0.001	3.41 (2.98, 3.84)	<0.001	2.83 (2.34, 3.33)	<0.001
Georgia (vs. Armenia)	−0.13 (−0.32, 0.06)	0.187	−0.95 (−1.21, −0.69)	<0.001	−0.46 (−0.66, −0.27)	<0.001	−0.82 (−1.06, −0.58)	<0.001
*Sociodemographics*								
Age	0.002 (−0.001, 0.004)	0.176	0.001 (−0.003, 0.005)	0.678	−0.01 (−0.01, −0.001)	0.026	0.00 (0.00, 0.01)	0.639
Female (vs. male)	0.06 (−0.05, 0.16)	0.289	0.14 (−0.01, 0.29)	0.074	−0.02 (−0.32, 0.29)	0.900	−0.29 (−0.64, 0.06)	0.105
Unemployed/other (vs. employed)	0.03 (−0.04, 0.10)	0.379	0.001 (−0.09, 0.10)	0.980	0.04 (−0.13, 0.21)	0.630	−0.01 (−0.20, 0.18)	0.929
Other (vs. married/cohabitating)	−0.05 (−0.12, 0.03)	0.262	−0.11 (−0.22, 0.01)	0.065	−0.09 (−0.29, 0.10)	0.340	0.16 (−0.06, 0.38)	0.150
Children <18 in home (vs. no)	0.06 (−0.1, 0.13)	0.118	0.02 (−0.08. 0.13)	0.647	−0.13 (−0.3, 0.03)	0.112	0.00 (−0.19, 0.19)	0.986
*Smoking Characteristics*								
Former smoker (vs. never)	−0.01 (−0.15, 0.13)	0.846	−0.05 (−0.25, 0.15)	0.622	-	-	-	-
Smoking some days (vs. every)	-	-	-	-	0.31 (0.04, 0.58)	0.022	0.51 (0.21, 0.81)	0.001
Importance of quitting	-	-	-	-	0.05 (0.02, 0.07)	<0.001	0.01 (−0.01, 0.04)	0.345
Confidence in quitting	-	-	-	-	−0.01 (−0.03, 0.02)	0.462	−0.02 (−0.05, 0.01)	0.117
Unconditional model ICC	17.79%		39.73%		13.90%		25.5%	
full model ICC	17.93%		16.49%		3.82%		5.7%

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
