# Peer review of "Smokers’ and Nonsmokers’ Receptivity to Smoke-Free Policies and Pro- and Anti-Policy Messaging in Armenia and Georgia"

_ijerph, 2020, doi:10.3390/ijerph17155527_

Round 1
Reviewer 1 Report
This important articles, with wide-view for tobacco products, will be a great help to understand difficulties and hopes in progress of tobacco control. If possible, please repeat or keep on monitoring general population over time, especially newer tobacco products or electric cigarettes.
Author Response
This important articles, with wide-view for tobacco products, will be a great help to understand difficulties and hopes in progress of tobacco control. If possible, please repeat or keep on monitoring general population over time, especially newer tobacco products or electric cigarettes.
Response: Thank you for this favorable review. We will ensure follow-up assessments and dissemination of findings.

Reviewer 2 Report
Overall, this is a meritorious paper although hardly exciting or ground-breaking. It's scientifically sound and meticulously sets out the research design, findings, and discussion. The findings are similar to what has been found in other settings on similar questions. What I would like to see is more 'big picture' discussion of how/why we might expect these two countries to be similar/different and what the patterns of significant findings indicate about these expectations, and why.
Author Response
Thank you for this feedback. We have included some higher-level comments addressing you ‘big picture’ suggestion.
- In the abstract, we have inserted: “Armenians versus Georgians generally perceived pro- and anti-policy messaging more persuasive.”
- Last paragraph of the introduction, we have revised to say: “Given the recent progressiveness of tobacco control in Georgia relative to Armenia, we anticipate greater support for such policies and persuasiveness of pro-policy messaging among Georgians versus Armenians. Additionally, we hypothesize that previously identified correlates of support for smoke-free policies and related messaging (e.g., female, nonsmoker) will hold in these populations, that policies in certain contexts (e.g., healthcare settings) will receive more support than in other settings (e.g., outdoor areas), and that some messaging strategies (e.g., health, individual rights) will be particularly persuasive.”
- We have written in the Discussion section: “This study supported some hypotheses (e.g., sociodemographic and tobacco use factors associated with policy support and perceived persuasiveness of messaging), but not others. Notably, while Armenians versus Georgians reported less support for policies (on average) in several settings, country of residence was not a significant correlate of policy support in multivariable analysis. In addition, Armenians versus Georgians generally indicated that both pro- and anti-policy messaging was more persuasive. Below, we further discuss, contextualize, and interpret these and other findings….”
“….Interestingly, Armenians were more likely to perceive both pro-policy and anti-policy messaging as more persuasive; this may be related to the fact that Georgia had (and has) made more progress in advancing smoke-free policies, which involved public health campaigns to garner such support. Perhaps the information provided to Armenians was perceived as more novel and thus more persuasive [39], given a relative dearth of such public health campaigns.”
Reviewer 3 Report
This is a very interesting study analyzing the receptivity to smoke-free policies and perceived persuasiveness of pro- and anti-policy messaging among 1,456 individuals in Armenia and Georgia. The results showed that, overall, participants in the study were more receptive to a cigarette smoke-free policy than for ENDs/HTPs, especially in health care, religious, government, and workplace settings; public transport; schools; and vehicles with children. However, they were less receptive of smoking bans in bars and restaurants. In general, participants felt more persuaded by pro-policy messages than anti-policy messages.
Smoking rates are dramatically high in these two countries. Thus, it is critical to understand how and what type of measures should be implemented to reduce tobacco burden in Armenia and Georgia. The manuscript addresses an important issue, it is well written, it includes a big sample size, and the results have important implications for policy and practice. However, there a few concerns regarding the lack of connection between the study aims and conclusions, lack of details in some aspects of the methods, and questions about how the variables were organized and analyzed. Below are some specific comments and questions:
- The introduction is very compelling and well written. The authors indicate that “a 2014 study documented similar findings in Georgia, such that nonsmokers, women, and those older showed more smoke-free policy support; however, this study did not assess attitudes toward ENDs/HTPs policies.” (pg. 2, lines 80-82). If the relevance and novelty of this study is in the data about ENDs/HTPs, then you should consider emphasizing this more throughout the introduction. For example, what is the prevalence of use of ENDs/HTP in Georgia and Armenia? Electronic cigarette use among youth is briefly mentioned, however this study is conducted among adults. We know some countries like the UK are more permissive or pro-e-cigarettes, but others like Australia are have restrictive laws about e-cigarette sales and use. What is the position of Georgia and Armenia regarding ENDs/HTPs?
- The aims of the study are not consistent throughout the manuscript. For example, in the abstract: “we examined receptivity to cigarette and electronic nicotine delivery systems (ENDs)/heated tobacco product (HTP) smoke-free policies in various locations and persuasiveness of pro- and anti-policy messaging,” but in the introduction “this study examined 1) nonsmokers’ and smokers’ receptivity to smoke-free policies, both with regard to cigarettes and ENDS/HTPs, across various locations; and 2) persuasiveness of messaging in support and opposition of such policies using 2018 survey data from adults in 28 cities in Armenia and Georgia.” This gives the sense that the aim is to compare smokers and non-smokers. Was the comparison of smokers vs non-smokers an exploratory analysis or the main question of the study? Should this be a secondary aim? This is even more confusing in the results section, in which all it reports is the difference between smokers and non-smokers.
- I understand that more detailed information about the parent study can be found in a previous publication. However, there is important information that should be summarized here, such as inclusion and exclusion criteria or the study procedures (how were participants approached, how did they respond to measures [eg., online, by phone])?
- How were importance and confidence in quitting assessed? Were reliable and validated questionnaires used?
- Were there any questions about use of ENDs/HTPs included?
- Please be consistent with the terms. Do “reactions to smoke-free policy” (methods) measure “receptivity” (aims)?
- It seems that the differences, although significant, are very small. For example, in table 2, in healthcare facilities non-smokers have a mean score of 4.89 and smokers of 4.74, which is very close, but with a p<.001. I wonder if the differences are due to the big sample size. Please consider using some type of correction or report the effect size.
- The means of the different scales assessing reactions and persuasiveness seem very high. Do they have a normal distribution? Reporting the percentage of participants who chose each category would seem more adequate than using means and standard deviations. Also, it seems that “don’t know” was considered as the middle because many people reported this option. Is this a good justification to place it in the middle? What happened with those who answered “refuse”?
- Are differences reported in Figure 1 significant?
- I think the figures should be placed in the results section rather than methods.
Round 2
Reviewer 3 Report
I appreciate the authors being responsive to all my comments. I have no other questions or concerns regarding this manuscript.